# Influence of Age and Harvesting Season on The Tensile Strength of Bamboo-Fibre-Reinforced Epoxy Composites

**DOI:** 10.3390/ma15124144

**Published:** 2022-06-10

**Authors:** Yalew Dessalegn, Balkeshwar Singh, Aart W. van Vuure, Ali A. Rajhi, Gulam Mohammed Sayeed Ahmed, Nazia Hossain

**Affiliations:** 1Program of Mechanical Design and Manufacturing Engineering, Department of Mechanical Engineering, Adama Science and Technology University, Adama 1888, Ethiopia; yalewdesu@yahoo.com (Y.D.); gmsayeed.ahmed@astu.edu.et (G.M.S.A.); 2Composite Materials Group, Department Materials Engineering, Campus Group T, KU Leuven, Andreas Vesaliusstraat 13, 3000 Leuven, Belgium; aartwillem.vanuure@kuleuven.be; 3Mechanical Engineering Department, College of Engineering, King Khalid University, Abha 61421, Asir, Saudi Arabia; arajhi@kku.edu.sa; 4Program of Mechanical Design and Manufacturing Engineering, Center of Excellence (COE) for Advanced Manufacturing Engineering, Adama Science and Technology University, Adama 1888, Ethiopia; 5School of Engineering, RMIT University, Melbourne, VIC 3001, Australia; bristy808.nh@gmail.com

**Keywords:** bamboo fibres, bamboo ages, epoxy composite, harvesting season, tensile strength, Young’s modulus

## Abstract

The purpose of this study was to measure the strength of various bamboo fibres and their epoxy composites based on the bamboo ages and harvesting seasons. Three representative samples of 1–3-year-old bamboo plants were collected in November and February. Bamboo fibres and their epoxy composites had the highest tensile strength and Young’s modulus at 2 years old and in November. The back-calculated tensile strengths using the “rule of mixture” of Injibara, Kombolcha, and Mekaneselam bamboo-fibre-reinforced epoxy composites were 548 ± 40–422 ± 33 MPa, 496 ± 16–339 ± 30 MPa, and 541 ± 21–399 ± 55 MPa, whereas the back-calculated Young’s moduli using the “rule of mixture” were 48 ± 5–37 ± 3 GPa, 36 ± 4–25 ± 3 GPa, and 44 ± 2–40 ± 2 GPa, respectively. The tensile strengths of the Injibara, Kombolcha, and Mekaneselam bamboo-fibre-reinforced epoxy composites were 227 ± 14–171 ± 22 MPa, 255 ± 18–129 ± 15 MPa, and 206 ± 19–151 ± 11 MPa, whereas Young’s moduli were 21 ± 2.9–16 ± 4.24 GPa, 18 ± 0.8–11 ± 0.51 GPa, and 18 ± 0.85–16 ± 0.82 GPa respectively. The highest to the lowest tensile strengths and Young’s moduli of bamboo fibres and their epoxy composites were Injibara, Mekaneselam, and Kombolcha, which were the local regional area names from these fibres were extracted. The intended functional application of the current research study is the automobile industries of headliners, which substitute the conventional materials of glass fibres.

## 1. Introduction

Bamboo plants are commonly referred to as “natural glass fibers” [1]. Not only do the extraction methods influence the mechanical properties of bamboo fibre but also the bamboo species, plant age, growing conditions, and the part of the culm from which the fibres are extracted. Numerous studies on tensile properties based on culm age yielded inconsistent results [2]. Bamboo fibres are acceptable for use as reinforcement in composite materials to produce light and strong composites, and they may be suitable for structural and semi-structural composite applications [3]. Several studies on bamboo fibre-reinforced composites, primarily with random technical fibre configuration, that used thermoset, thermoplastic, and biodegradable matrices were published [4]. Many factors can influence the characteristics of elementary bamboo fibres, including the bamboo species, fibre location within the culm, and the age of the culm. According to previous research, sympodial bamboos have a lot of potential in terms of their properties. The mean Young’s moduli of Ma bamboo (Dendrocalamus latiflorus Munro) and Moso bamboo (Phyllostachys edulis) are 45.8 GPa and 34.6 GPa, respectively, but they have a similar tensile strength at 4 years. The Young’s modulus, as well as strength, of Moso bamboo fibres varied very little across samples at the ages of 0.5 to 4 years. After six months, the mechanical properties of the cell wall have little effect. Moreover, at the ages of 0.5–8.5 years, the mean of Young’s moduli and strain to failure of Moso bamboo fibres show no significant differences between the group. Bamboo ages have a significant effect on tensile properties. However, only a small number of studies on the influence of ageing on mechanical strength were conducted [5]. The tensile strength of bamboo was recorded at 250 MPa or more, depending on the location, type of species, and cross-sectional area [6]. The moisture content is influenced by the harvesting season of the bamboo culm, which reaches a minimum at the end of the dry season and the highest during the showery season. According to previous studies of the four Phyllostachys species grown in Ireland, in the spring, they have much less moisture than in the summer [7]. Bamboo characterisation in terms of anatomical, physical, mechanical, and chemical properties aids in determining the maturation age, which improves processing and utilisation [8]. Bamboo culms reach their maximum strength after two to three years of maturation [9]. Bamboo properties, on the other hand, vary depending on the species, age, location, and external factors [10]. According to Correal and Arbelaez [11,12], bamboo’s age has a significant impact on its mechanical behavior. Fibre length, orientation, and fibre volume fraction influence the mechanical strength of the composites. When a composite is subjected to a load, the stress is transmitted along the fibres. When comparing mechanical properties, long fibres outperformed short fibres [13]. The advantages of bamboo plants are the reduced carbon footprint, low price, reduction in erosion, fast growth rate, and the capability to fix CO_2_ in the atmosphere [14]. The properties of bamboo fibres should be several times stronger than the matrix to make a considerable difference in the final composite’s properties. The properties of bamboo fibres are their low density, renewable, abundant, non-abrasive, non-corrosive, biodegradable, low price, higher specific strength, stiffness, and low environmental impact throughout their entire life cycle [15]. The main drawbacks of natural fibres are their hydrophilic nature, restricted thermal stability, and poor adhesion with adjacent counterparts [16]. Nowadays, researchers have developed natural fibres and their polymer composites to reduce the environmental impact and satisfy the economic problem of the community with a positive impact on composite sustainability. Natural fibres have a low environmental effect during material processing, the usage phase, and salvage compared with synthetic fibres [17]. Researchers demonstrated the ability of natural fibres to practically replace the glass fibres commonly utilised as reinforcement in composites by producing hybrid composites. Hence, natural fibres offer an environmentally friendly and cost-effective alternative to existing composites [18]. The most difficult aspect of working with natural-fibre-reinforced composites (NFRCs) is the wide range of properties and characteristics. The properties of NFRCs are influenced by several factors, including the fibre type, environmental conditions, processing methods, and fibre modification [19]. The mechanical behaviour of bamboo fibres is required to improve developments in bamboo fibre polymer composites, which will require data on elementary fibre properties. While many other investigations were conducted in this area, the majority of them concentrated on mechanical differences at the single-fibre scale concerning age or location in bamboo culms [20]. The bamboo plant is the fastest-growing and most renewable plant that reaches its full growth in just a few years because the maturity cycle of bamboo is 3 to 4 years long. Even though the utilisation potential of bamboo for several applications has been applied, much greater specific mechanical properties have not been effectively utilised for polymer-based composites. Bamboo fibres have comparable mechanical properties to sisal, vakka, and banana, but they have higher tensile and flexural properties compared to other natural fibres [21]. Bamboo composites are used in military, automobile, chemical, industrial, electrical, hydraulic, nautical, and aerospace applications due to their ability to maintain a sustainable environment, design flexibility, and continuous improvement in the face of a variety of challenges [22]. Nowadays, a hybrid of carbon with glass-fibre-reinforced polymer composites is produced to improve the mechanical properties and reduce the cost of the materials. The two types of fibre hybridisation are intra-yarn hybrid (uniform dispersed hybrid (UDH)) and inter-layer hybrid (core–shell hybrid (CSH)). The maximum percentages of available strength increased by uniformly dispersing the carbon fibre into the glass fibre were up to 10.9% for short beam shear, 60.3% for three-point bending, and 58.7% for tensile strength, respectively [23]. This study aimed to measure the strength of bamboo fibres and their epoxy composites based on Ethiopian bamboo species, ages, and harvesting seasons. The aim of the current study was to substitute the conventional headliner products of glass fibres with bamboo fibres. The advantages of glass fibres are their high tensile strength, good thermal resistance, and good resistance to moisture. However, they are not biodegradable, require high production energy, and have a high density. Previously, several researchers studied various parameters of bamboo, including their tensile properties. Over 1 million hectares are covered with bamboo plants in Ethiopia, but they are used for furniture making, building houses, and fencing, where they are used under their utilisation capacity. Nowadays, natural fibres are substituted by glass fibres that are non-biodegradable; high density; high cost; utilise high energy for production; and are environmentally unfriendly during the production, usage, and salvage phase. The problems with synthetic fibres are solved by the investigation of natural fibres (bamboo fibres), which are substituted for synthetic fibres. Bamboo fibres have higher properties compared with other natural fibres. However, insufficient literature can be found on the tensile strength, Young’s modulus, and strain to failure of bamboo fibre and their epoxy composite obtained from Ethiopia. The current study was motivated by the growing needs for lightweight, cost-effective, and ecologically friendly composites for industrial applications.

## 2. Materials and Methods

### 2.1. Studies Area Description

Bamboo species were harvested from three regions of Ethiopia (Injibara, Kombolch, and Mekaneselam town). The types of bamboo species found in the Injibara and Mekaneselam regions were Yusania alpina, whereas the Kombolcha region had Bambusa oldhamii. Injibara bamboo was grown in the Amhara state, West Gojam Zone, near Injibara. It is currently located in the southwestern region of the state and the northwestern region of the country, Ethiopia. It is 447 km from Ethiopia’s capital city, Addis Ababa, and 118 km from Bahir. The annual relative humidity and temperature of Injibara are 64% and 22 °C, respectively, and its altitude is 2560 m [24]. The colour of the Injibara bamboo culm changes from green to black as the age of the bamboo increases. Kombolcha is found in the South Wollo Zone of Ethiopia’s Amhara state, 13 km from Dessie City. The colour of the bamboo culm changes from dark green to light green as the age of the bamboo increases. The annual relative humidity and temperature of Kombolcha are 56% and 25.8 °C, respectively, and its altitude is 1857 m [25,26,27]. Makenaselam is located in the South Wollo Zone of Ethiopia’s Amhara state, 180 km from Dessie City. The colour of the bamboo culm changes from dark green to bright green as the age of the bamboo increases. The annual relative humidity and temperature of Mekaneselam are 65% and 20 °C, respectively, and its altitude is 2605 m [28,29,30].

### 2.2. Sampling Technique

As shown in Figure 1, the samples were harvested in three regions and two seasons in Ethiopia. Moreover, three representatives of bamboo species were harvested at the ages of 1, 2, and 3 years. Experienced field personnel determined the age of the bamboo culm based on its colour and sheath. The culm height was divided into three main sections: the bottom, middle, and top parts. The extracted bamboo fibre is found in the middle part of the culm because of the average diameter, thickness, and larger intermodal length that are found in the middle parts of the bamboo culm.

### 2.3. Microscopic Observations

Optical microscopic observations on transverse sections of the culm were made to investigate the distribution of the vascular bundles, as well as the microstructure of the fibres. The samples (1 cm^3^) were cut with a sledge microtome (Reichert, Vienna, Austria) to minimise unwanted impurities that can appear after grinding or polishing techniques [31,32,33]. The samples were embedded using Buehler’s EpoxiCure 2 epoxy mounting resin and EpoxiCure 2 hardener. The samples were sanded and polished before being examined under an optical microscope (Axioskop 40, Zeiss, Jena, Germany) to determine the ratio of fibre bundles.

### 2.4. Composite Production

The UD bamboo fibre epoxy composites were made using compression moulding. Weight measurements were used to confirm a 40% fibre volume fraction. The fibres were dried in the oven at 60 °C for 24 h, then cut to a 25 cm length and weighted based on the volume fraction. The fibres were impregnated and vacuum filmed inside the mould cavity to produce a unidirectional composite see Figure 2a–c. Degassed resin was used to impregnate the fibres and they were poured onto the fibres inside the mould. The entire mould was inserted into the hot press at 75 °C for 1 h for consolidation, removed by demoulding, and post-curing was performed at 150 °C for 1 h. After post-curing, the specimens were taken out of the oven and prepared for testing at 21 ± 2 °C and 50% RH. The moulding process is indicated in Figure 2, which is a modernized version of the setup used by [34,35], and the samples were geometrically comparable. Since the fibres’ weight in the mould was controllable, spacers were used to control the specimen thickness and manage the fibre volume fractions, as illustrated in Figure 2c. The mould’s width determined the size of the specimen. Therefore, the specimens were produced as a unidirectional composite with small cross-sections, which typically resulted in a lower number of defects and greater control over the fibre orientation [36,37,38].

### 2.5. Determination of Fibre Volume Fraction

The fibre volume fraction (Vf) was calculated in the following manner and following the ISO 14127:2008 standard [39]:(1)$$\mathrm{Vf}=\frac{\mathrm{mf}/\rho f}{\mathrm{Vc}}$$
where mf denotes the mass of the fibres (mf = vf × Vc × ρ_f_), Vc denotes the volume of the composite, and ρ_f_ denotes the density of the fibres.

### 2.6. Back-Calculation of The Fibre Properties

The strength and stiffness of the impregnated fibre bundles can be used to evaluate the density of bamboo fibres as they exist and behave in a composite. The fibre characteristics can be back-calculated from the measured composite properties of this impregnated fibre bundle using the inverted formulae of the well-known “rule of mixture”:(2)$$\mathrm{Ef}=\frac{\mathrm{Ec}-\mathrm{Em}\;*\;\left(1-\mathrm{Vf}\right)}{\mathrm{Vf}}$$
(3)$$\sigma f=\frac{\sigma c-\sigma^{\prime }m\;*\;\left(1-\mathrm{Vf}\right)}{\mathrm{Vf}}$$
where Ef denotes the modulus of the fibre; Em denotes the matrix modulus; Vf denotes the volume fraction of the fibre; σf denotes the strength of the fibre; and σ^′^m denotes the strength of the matrix, which can be calculated by considering linear elastic matrix deformation up to the moment of the fibre (and thus composite) failure. Equation (4) is only acceptable when the fibres fail first, implying that they do have a lower failure strain than the matrix.
(4)$$\sigma^{\prime }m=\mathrm{Em}\;*\;\varepsilon u,\;c.$$

### 2.7. Preparation of Resin

Before the production and measurement, the bamboo fibres were dried for four days at 60 °C. As a matrix, Hexion’s Epikote 828 LVEL with a 1,2-diaminocyclohexane (Dytek DCH-99) hardener was applied. The epoxy-to-hardener relation was 100:15.2. For 10 min, the mixture resin was degassed in a vacuum oven. Pre-curing took place at 75 °C for 1 h, then 1 h of post-curing at 150 °C in the oven. Composites measuring 250 × 10 × 2 mm were produced with a volume fraction of 40% fibres measured depending on the weight and density of the fibres. The tensile strength, stiffness, and strain to failure of the epoxy matrix were 70 MPa, 2.7 GPa, and 4.1%, respectively [40].

### 2.8. Extraction of Bamboo Fibres Using A Rolling Machine

Bamboo culm was cut from the nodes across a cross-section, and then the internodes were split along the longitudinal axis into strips of 30 mm. One-third of the inner and outer parts of the culm were removed after enough rolling, and then the middle part of the rolled culm was combed using different sizes of comb for separate fibres. For two weeks, the obtained fibres were sun-dried, and their lengths ranged above 250 mm [41]. As indicated in Figure 3a–c, internodal numbers 9 and 13 were used for the fibre extraction because they were found at the middle part of the culm. The extraction of fibre occurred during the time of the harvest week. The cleaning was carried out using various sizes of comb with water added to remove the residual parenchyma. Finally, the fibres were dried at 60 °C for at least one week after extraction before being used to make the composites.

### 2.9. Bundle Test of Impregnated Fibre

Impregnated fibre bundle (IFBT) tests were developed and the stiffness and strength of unidirectional composites were determined using the “rule of mixtures” and Equns. 5 and 6. Ef and Em denote the fibre and matrix moduli, respectively; 𝜎f and 𝜎^′^m denote the fibre and matrix tensile strengths, respectively, and Vf and Vm denote the fibre and matrix volume fractions, respectively.
(5)Ec=VfEf+VmEm
(6)$$\sigma c=\mathrm{Vf}\sigma f+\mathrm{Vm}\sigma^{\prime }m$$

### 2.10. Tensile Test

Specimens were prepared under ASTM D3039 [42]. For the tests, a model 4467 machine with a 30 kN load cell and a crosshead speed of 2 mm/min was applied. The distance between the two ends of the jaws was positioned at 150 mm, and an extensometer with a length of 25 mm was applied to accurately measure the elongation of the composites. Because there was no failure close to the clamps, end tabs were not required; instead, the specimens wemechanically clamped with sandpaper in the grips to prevent slippage. Figure 4 shows the tensile experimental setup. Before testing, all specimens were conditioned for at least 24 h at a temperature of 21 ± 2 °C and 50% RH.

### 2.11. Statistical Analysis

Stata 12.0 was used for the statistical analysis. The data were tabulated and the significance of the difference between the mean values of the various groups was determined using one-way ANOVA. The difference in mean values in impact strength, tensile strength, Young’s modulus, and strain to failure between the different groups of bamboo species was determined using pairwise comparisons of means [43].

## 3. Results and Discussions

### 3.1. Tensile Strength Measurement

The influences of age and harvest season on the back-calculated tensile strength of bamboo species in Ethiopia are shown in Figure 5. The highest and lowest back-calculated tensile strengths of Injibara (*Y. alpina*) were 539 ± 74 MPa and 422 ± 33 MPa were recorded at 2 and 3 years old in February, whereas in November, 548 ± 40 MPa and 431 ± 92 MPa were recorded at 2 and 1 year old, respectively. The back-calculated tensile strength of November had 1.6% higher than in February for Injibara bamboo (*Y. alpina*). The highest and lowest back-calculated tensile strengths of Kombolcha bamboo (*B. oldhamii*) were 449 ± 26 MPa and 339 ± 30 MPa and were recorded at 2 and 3 years old in February, whereas in November, 496 ± 16 MPa and 422 ± 45 MPa were recorded at 2 and 3 year old, respectively. The back-calculated tensile strength in November was 9.5% higher than in February for Kombolcha bamboo (*B. oldhamii*). The highest and lowest back-calculated tensile strengths of Mekaneselam bamboo (*Y. alpina*) were 507 ± 24 MPa and 421 ± 55 MPa and were recorded at 2 and 3 years old in February, whereas in November, 541 ± 21 MPa and 399 ± 55 MPa, respectively, were recorded at 2 and 3 years old. The Mekaneselam bamboo harvested in November had a 6.19% higher back-calculated tensile strength than that harvested in February. The highest back-calculated tensile strengths of the Injibara (*Y. alpina*) bamboo fibres (548 ± 40 MPa) were higher by 12%, 20%, and 40%, whereas the Kombolcha (*B. oldhamii*) bamboo fibres (496 ± 16 MPa) were higher by 3%, 12%, and 34%, and the Mekaneselam (*Y. alpina*) bamboo fibres (541 ± 21 MPa) were higher by 11%, 19%, and 39% than Guadua Angustifolia Kunth (GAK) (481 MPa), Dendrocalamus membranaceus Munro (DMM) (436 MPa), and phyllostachys nigra Boryana (PNB) (329 MPa), respectively [44]. From the highest to the lowest, the calculated tensile strengths were recorded in Injibara (*Y. alpina*), Mekaneselam (*Y. alpina*), and Kombolcha (*B. oldhamii*). The current research investigation showed that 2-year-old bamboo harvested in November recorded the highest back-calculated tensile strength.

### 3.2. Young’s Modulus Measurement

As shown in Figure 6, the highest and lowest back-calculated Young’s moduli of the Injibara (*Y. alpina*) bamboo fibres were 44 ± 4 GPa and 37 ± 3 GPa and were recorded at 2 and 3 years old in February, whereas in November, 48 ± 5 GPa and 37 ± 5 GPa were recorded at 2 and 1 year old, respectively. November had an 8% higher Young’s modulus than February for the Injibara (*Y. alpina*) bamboo fibres. The highest and lowest back-calculated Young’s moduli of the Kombolcha (*B. oldhamii*) bamboo fibres were 33 ± 2 GPa and 25 ± 4 GPa and were recorded at 2 and 3 years old in February, whereas in November, 36 ± 4 GPa and 29 ± 5 GPa were recorded at 2 and 3 years old, respectively. November had an 8% higher back-calculated Young’s modulus than in February for the Kombolcha (*B. oldhamii*) bamboo fibres. The highest and lowest back-calculated Young’s modulus of the Mekaneselam (*Y. alpina*) bamboo fibres of 43 ± 2 GPa and 40 ± 3 GPa were recorded at 2 and 3 years old in February, whereas in November, 44 ± 2 GPa and 40 ± 2 GPa were recorded at 2 and 3 years old, respectively. November had a 2.27% higher back-calculated Young’s modulus than February for the Mekaneselam bamboo fibres. The highest to the lowest back-calculated Young’s moduli were recorded in Injibara (*Y. alpina*), Mekaneselam (*Y. alpina*), and Kombolcha (*B. oldhamii*), respectively. The current research investigation showed that 2-year-old bamboo harvested in November recorded the highest back-calculated Young’s modulus. The highest back-calculated Young’s moduli of the Injibara (*Y. alpina*) bamboo fibres (48 ± 5 GPa) was higher by 8%, 15%, and 23%, whereas the Kombolcha (*B. oldhamii*) bamboo fibres (36 ± 4 GPa) were lower by 18%, 12%, and 3%, and the Mekaneselam (*Y. alpina*) bamboo fibres (44 ± 2 GPa) had a similar Young’s modulus to GAK (44 GPa), and higher by 7%, and 16% than DMM (41 GPa) and PNB (37 GPa), respectively.

### 3.3. Strain to Failure Measurement

As shown in Figure 7, the highest and lowest strains to failure of the Injibara (*Y. alpina*) bamboo fibre composites were 1.2 ± 0.1% and 1.07 ± 0.07% and were recorded at 2 and 3 years old in February, whereas in November, 1.08 ± 0.08% and 1.04 ± 0.12% were recorded at 2 and 3 years old, respectively. The Injibara bamboo (*Y. alpina*) harvested in February was 3.7% more susceptible to failure than that harvested in November. The highest and lowest strains to failure of Kombolcha (*B. oldhamii*) were 1.35 ± 0.16% and 1.16 ± 0.07% and were recorded at 2 and 3 years old in February, whereas in November, 1.17 ± 0.03% and 1.14 ± 0.03% were recorded at 1 and 2 years old, respectively. The Kombolcha (*B. oldhamii*) harvested in February had a 13.33% higher strain to failure than that harvested in November. The highest and lowest strains to failure of the Mekaneselam (*Y. alpina*) bamboo fibre composites were 1.07 ± 0.07% and 0.98 ± 0.07% and were recorded at 2 and 3 years old in February, whereas in November, 1.13 ± 0.075% and 0.92 ± 0.09% were recorded at 1 and 3 years old, respectively. The Mekaneselam harvested in November had a 5.31% higher strain to failure than that cut in February. All Ethiopian bamboo species harvested in February were more susceptible to failure compared to those harvested in November. The strain to failure of the Injibara bamboo fibre composites (1.08%) was comparable with the previous studies of GAK (1.06%) but was higher than those of DMM (0.93%) and PNB (0.77%), whereas the Kombolcha bamboo fibre composites (1.14%) had higher strain to failure than the previous studies of GAK (1.06%), DMM (0.93%), and PNB (0.77%); moreover, the Mekaneselam bamboo fibre composites (1.11%) were comparable with the previous studies of GAK (1.06%) but higher than DMM (0.93%) and PNB (0.77%) [45].

### 3.4. Young’s Moduli of Composites Using IFBTs

As *indicated* in Figure 8, the *Young’s* moduli of composites for the Injibara bamboo (*Y. alpina*), Kombolcha bamboo (*B. oldhamii*), and Mekaneselam bamboo (*Y. alpina*) were measured experimentally. The highest Young’s modulus of the Injibara bamboo fibre epoxy composites of 19 ± 0.72 GPa and 18 ± 0.9 GPa were recorded as the theoretical and experimental values, respectively, in February at the age of 2 years old. An efficiency factor of 95% was reached after comparing the experimental value (18 GPa) with the theoretical value (19 GPa) in February for the Injibara bamboo (*Y. alpina*). In November, Young’s modulus of the Injibara bamboo fibre composites of 21 ± 1.1 GPa and 20 ± 0.92 GPa were recorded as the theoretical and experimental values, respectively, at 2 years old. An efficiency factor of 95% was reached after comparing the experimental value (18 GPa) with the theoretical value (19 GPa) in November for the Injibara bamboo (*Y. alpina*). The highest theoretical and experimental values of Young’s modulus of the Kombolcha (*B. oldhamii*) bamboo fibre epoxy composites were 17 ± 0.82 GPa and 16 ± 1.1 GPa and were recorded in February, whereas in November, 18 ± 0.86 GPa and 17 ± 0.85 GPa were recorded at 2 years old, respectively. An efficiency factor of 94% was recorded in February and November for the Kombolch bamboo. As shown in Figure 8, the highest theoretical and experimental values of Young’s modulus of the Mekaneselam bamboo fibre composites were 19 ± 0.92 GPa and 18 ± 0.85 GPa and were recorded at 2 years old in February, whereas in November, 19 ± 0.95 GPa and 18 ± 0.86 GPa were recorded at 2 years old, respectively. An efficiency factor of 95% was recorded in February and November for the Mekaneselam bamboo. In general, the theoretical value of Young’s modulus of the Ethiopian bamboo fibre epoxy composites was comparable with the experimental value. This indicated that the samples of the composites were prepared precisely and had good fibre/matrix adhesion.

### 3.5. Ultimate Tensile Strength of Bamboo Fibre Composites

The influence of age and harvest season on the ultimate tensile strengths of the bamboo fibre epoxy composites are shown in Figure 9. The highest and lowest ultimate tensile strengths of the Injibara (*Y. alpina*) bamboo fibre epoxy composites were 209 ± 23 MPa and 191 ± 25 MPa and were recorded at 2 and 3 years old in February, whereas in November, 227 ± 15 MPa and 171 ± 22 MPa were recorded at 2 and 1 year old, respectively. The ultimate tensile strength of the Injibara bamboo (*Y. alpina*) bamboo fibre epoxy composites was 9% higher with bamboo harvested in November than in February. The highest and lowest ultimate tensile strengths of the Kombolcha bamboo (*B. oldhamii*) bamboo fibre epoxy composites were 175 ± 32 MPa and 139 ± 21 MPa and were recorded at 2 and 3 years old in February, whereas in November, 188 ± 29 MPa and 129 ± 18 MPa were recorded at 2 and 3 years old, respectively. The ultimate tensile strength of the Kombolcha (*B. oldhamii*) bamboo fibres epoxy composites was 7% higher for the bamboo harvested in November than in February. The highest and lowest ultimate tensile strengths of the Mekaneselam (*Y. alpina*) bamboo fibre epoxy composites were 199 ± 34 MPa and 152 ± 22 MPa and were recorded at 2 and 3 years old in February, whereas in November, 206 ± 36 MPa and 166 ± 26 MPa were recorded at 2 and 3 years old, respectively. The Mekaneselam (*Y. alpina*) bamboo fibres epoxy composites had a 3% higher ultimate tensile strength for the bamboo harvested in November than in February.

### 3.6. Statistical Experimental Results of Tensile Strength

As presented in Table 1, the pairwise comparison test reveals that 2 compared with 1 year old and 3 compared to 2 years old were statistically significant; however, the age of 3 compared to 1 year old, as well as the harvesting months, were statistically insignificant at α = 0.05 for the tensile strength of the Injibara bamboo fibres.

**Table 1 materials-15-04144-t001:** Pairwise comparisons of the mean of the tensile strength for Injibara bamboo.

				Tukey’s Method			
Factors	UTS	Coeffi.	Std. Error	t	α>/t/	95% Confid. Interval	
Year	2 and 1	79.6	29.3649	2.71	0.03	6.79216	152.408
	3 and 1	−19.1	29.3649	−0.65	0.794	−91.908	53.7078
	3 and 2	−98.7	29.3649	−3.36	0.006	−171.51	−25.892
Month	2 and 11	−3.4	28.5467	−0.12	0.906	−61.875	55.0753

Year: 1, 2, and 3 are years. Months 2 and 11 are November and February, respectively. As presented in Table 2, the pairwise comparison test revealed that 2 compared with 1 and 3 compared with 2 years old were statistically significant; however, 3 compared with 1 year old, as well as the harvesting months, were statistically insignificant at α = 0.05 for Young’s modulus of the Injibara bamboo fibres.

As presented in Table 3, the pairwise comparison test revealed that 2 compared with 1 year old and 3 compared with 2 years old were statistically significant; however, 3 compared with 1 year old, as well as the harvesting months, were statistically insignificant at α = 0.05 for the tensile strength of the Kombolcha bamboo fibres.

### 3.7. Young’s Modulus of Kombolcha Bamboo Using ANOVA

As presented in Table 4, the pairwise comparison test revealed that 2 compared to 1 year old and 3 compared with 2 years old were statistically significant; however, 3 compared with 1 year old, as well as the harvesting months, were statistically insignificant at α = 0.05 for the tensile strengths of Kombolcha bamboo fibres.

As presented in Table 5, the pairwise comparison test revealed that 3 compared with 1 year old and 3 compared with 2 years old were statistically significant; however, 2 compared with 1 year old, as well as the harvesting months, were statistically insignificant at α = 0.05 for the tensile strengths of the Mekaneselam bamboo.

As presented in Table 6, the pairwise comparison tests revealed that 3 compared with 1 year old and 3 compared with 2 years old were statistically significant; however, 2 compared with 1 year old, as well as the harvesting months, were statistically insignificant at α = 0.05 for Young’s moduli of the Mekaneselam bamboo fibres.

## 4. Conclusions

The aim of this study was to measure the ultimate tensile strengths of Ethiopian bamboo fibres and their epoxy composites under the effects of age and harvesting season. The highest and the lowest tensile strengths and tensile moduli of the Kombolcha (*B. oldhamii*) and Mekaneselam (*Y. aplina*) bamboos were recorded at the ages of 2 and 3 years old in the harvesting seasons of February and November, respectively. In contrast, the highest and the lowest tensile strengths and tensile moduli of the Injibara bamboo were recorded at the ages of 2 and 3 years old in February. However, the highest and lowest tensile strengths and tensile moduli of the Injibara bamboo were recorded at the ages of 2 and 1 years old in November. The harvesting season of November produced higher mechanical properties compared with February. The highest to the lowest ultimate tensile strengths of the Ethiopian bamboo fibres and their epoxy composites were recorded as Injibara (*Y. alpina*), Mekaneselam (*Y. alpina*), and Kombolcha (*B. oldhamii*). The highest back-calculated tensile strengths of the Injibara (*Y. alpina*), Kombolcha (*B. oldhamii*), and Mekaneselam (*Y. alpina*) bamboo fibres were 548 ± 40 MPa, 496 ± 16 MPa, and 541 ± 41 MPa and were recorded at 2 years old in November, whereas in February, 539 ± 37 MPa, 449 ± 26 MPa, and 507 ± 24 MPa were recorded, respectively. The highest ultimate tensile strengths of the Injibara (*Y. alpina*), Kombolcha (*B. oldhamii*), and Mekaneselam (*Y. alpina*) bamboo fibre epoxy composites were 227 ± 15 MPa, 188 ± 29 MPa, and 206 ± 36 MPa and were recorded at 2 years old in November, whereas in February, 206 ± 23 MPa, 175 ± 32 MPa, and 199 ± 34 MPa were recorded, respectively. The highest back-calculated Young’s moduli of the Injibara (*Y. alpina*), Kombolcha (*B. oldhamii*), and Mekaneselam (*Y. alpina*) bamboo fibres were 48 ± 5 GPa, 36 ± 4 GPa, and 44 ± 2 GPa and were recorded at 2 years old in November, whereas in February, 44 ± 4 GPa, 33 ± 2 GPa, and 43 ± 2 GPa were recorded, respectively. The highest Young’s moduli of the Injibara (*Y. alpina*), Kombolcha (*B. oldhamii*), and Mekaneselam (*Y. alpina*) bamboo fibres epoxy composites were 20 ± 1.1 GPa, 18 ± 0.8 GPa, and 18 ± 0.86 GPa and were recorded at 2 years old in November, whereas in February, 18 ± 0.9 GPa, 17 ± 1.1 GPa, and 18 ± 0.85 GPa were recorded, respectively. The findings of this study can be fruitfully used to manufacture composites by bonding with appropriate matrixes to fabricate value-added products. According to the current research results, Ethiopian bamboo fibres and their epoxy composites should be utilised for the production of composite products, which replace glass fibre epoxy composites that are applied in automobile headliners.

## Figures and Tables

**Figure 1 materials-15-04144-f001:**
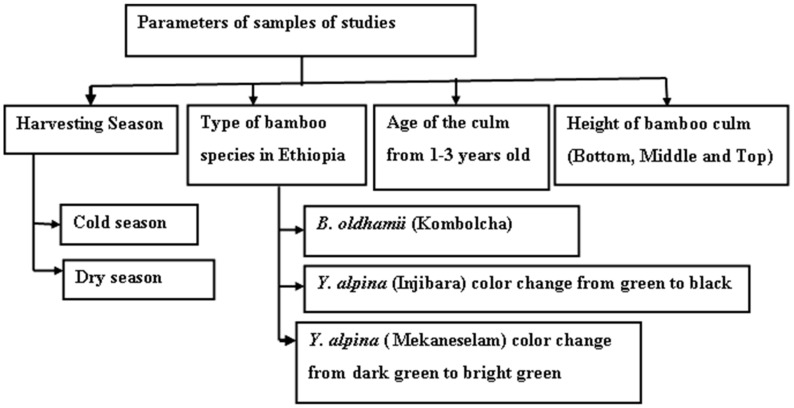
Sample-taking techniques of the bamboo culm.

**Figure 2 materials-15-04144-f002:**
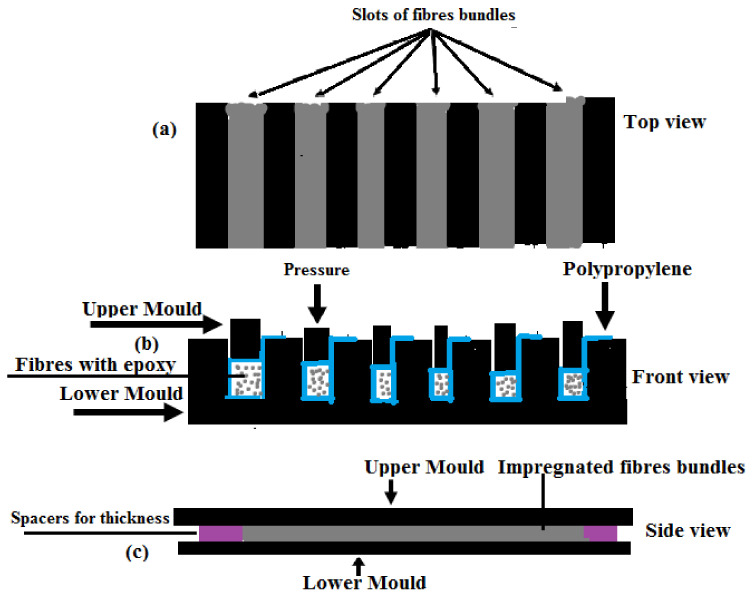
Testing of impregnated fibre bundles: (**a**) six mould cavities from the top view; (**b**) a front view of the mould’s cavities (spacer not placed, with the vacuum film surroundings all the bamboo fibre bundle); and (**c**) the counter-mould in the longitudinal cross-section with the spacers.

**Figure 3 materials-15-04144-f003:**
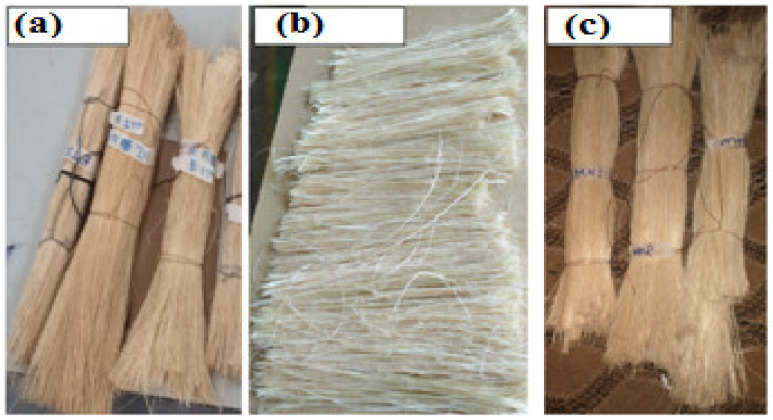
Extracted bamboo fibres: (**a**) Injibara, (**b**) Kombolcha, and (**c**) Mekaneselam.

**Figure 4 materials-15-04144-f004:**
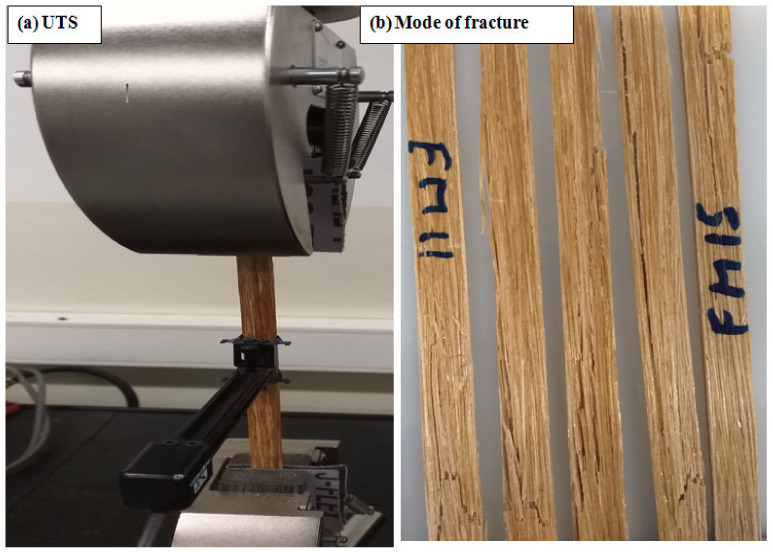
(**a**) Setup for the tensile test and (**b**) fracture after testing.

**Figure 5 materials-15-04144-f005:**
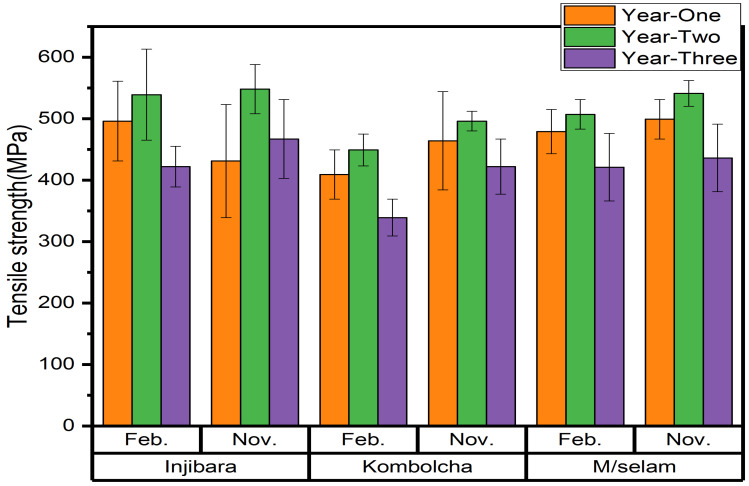
Back-calculated tensile strength of Ethiopian bamboo fibres using IFBTs.

**Figure 6 materials-15-04144-f006:**
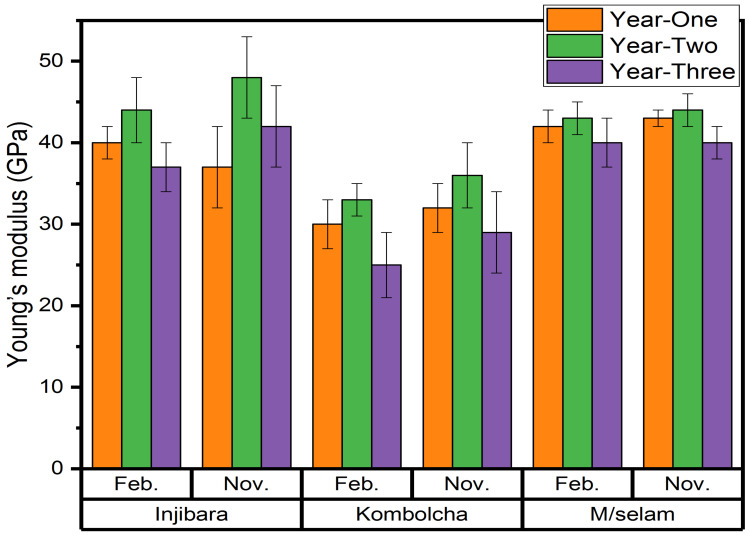
Back-calculated Young’s moduli using IFBTs.

**Figure 7 materials-15-04144-f007:**
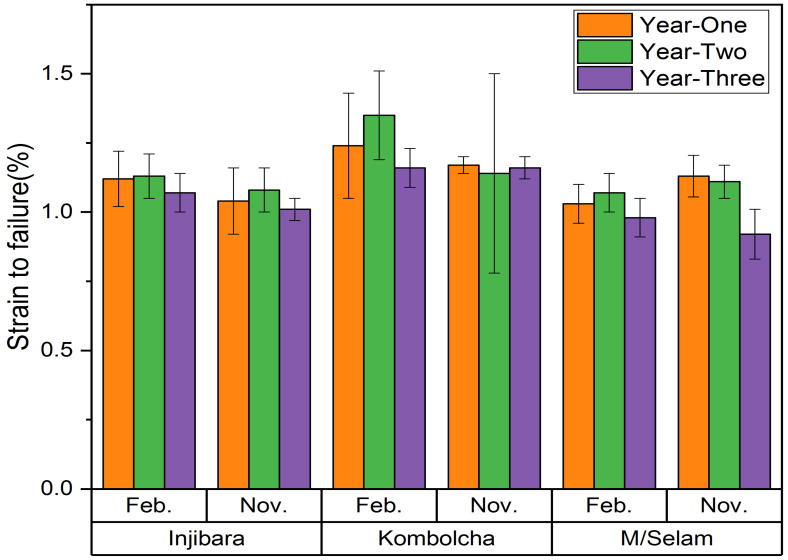
Strain to failure results using IFBTs.

**Figure 8 materials-15-04144-f008:**
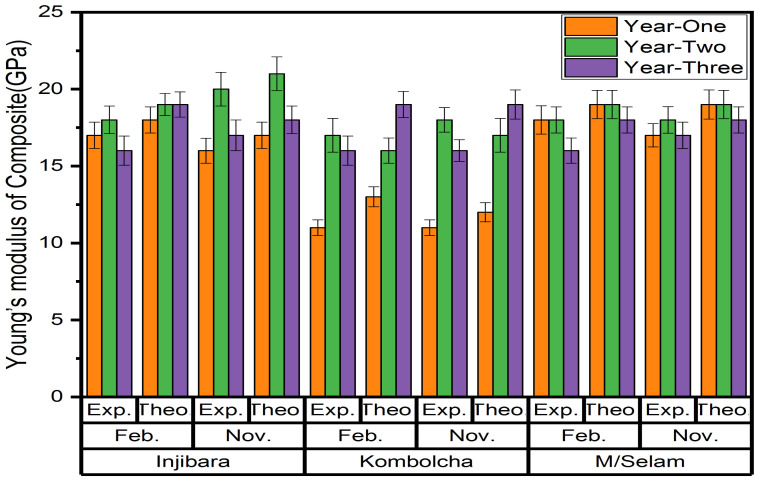
The experimental and theoretical values of Young’s moduli of the composites.

**Figure 9 materials-15-04144-f009:**
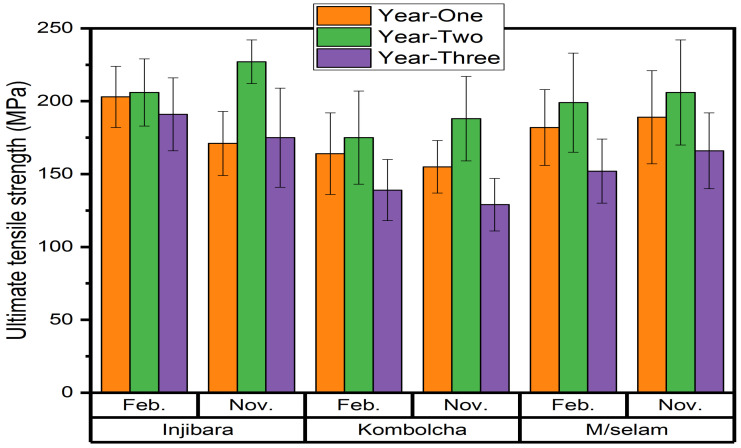
Ultimate tensile strengths of the bamboo fibres epoxy composites.

**Table 2 materials-15-04144-t002:** Pairwise comparisons of the mean of Young’s moduli for the Injibara bamboo.

Factors	E			Tukey’s Method			
		Coeffi.	Std. Error	t	α>/t/	95% Confid. Interval	
Years	2 and 1	7.1	2.19106	3.24	0.009	1.66745	12.5326
	3 and 1	0.6	2.191059	0.27	0.96	−4.83255	6.032551
	3 and 2	−6.5	2.191059	−2.97	0.017	−11.9326	−1.06745
Months	2 and 11	2.06667	2.10004	0.98	0.333	−2.2351	6.3684

**Table 3 materials-15-04144-t003:** Pairwise comparisons of the mean of the tensile strengths for the Kombolch bamboo.

Factors	UTS			Tukey’s Method			
		Coeffi.	Std. Error	t	α>/t/	95% Confid. Interval	
Ages (Yrs)	2 and 1	144	30.46	4.73	0.000	68.45	219.55
	3 and 1	129	30.46	4.23	0.001	53.45	204.55
	3 and 2	−15	30.46	−0.49	0.876	−90.55	60.55
Months	2 and 11	23.13	34.27	0.67	0.505	−47.069	93.34

**Table 4 materials-15-04144-t004:** Pairwise comparisons of the mean of Young’s moduli for the Kombolcha bamboo.

Factors	Compar.			Tukey’s Method			
		Coeffi.	Std. Error	t	>/t/	95% Confid. Interval	
Ages (Yrs)	2 and 1	11.3	2.93	3.86	0.002	4.033	18.57
	3 and 1	12.6	2.93	4.3	0.001	5.33	19.87
	3 and 2	1.3	2.93	0.44	0.898	-5.97	8.57
Months	2 and 11	6.47	2.93	2.2	0.036	0.459	12.475

**Table 5 materials-15-04144-t005:** Comparisons of tensile strengths for the Mekaneselam bamboo (*Y. alpina*) due to the effects of ages and harvesting months.

Factors	Comparison			Tukey’s Method			
		Coeffi.	Std. Error	t	α>/t/	95% Confid. Interval	
Ages (Yrs)	2 and 1	16.4	19.37	0.85	0.678	−31.63	64.43
	3 and 1	−97.7	19.37	−5.04	0.000	−145.73	−49.67
	3 and 2	−114.1	19.37	−5.89	0.000	−162.13	−66.07
Months	2 and 11	23.2	24.17	0.96	0.345	−26.32	72.72

**Table 6 materials-15-04144-t006:** Pairwise comparisons of the mean of Young’s moduli for the Mekaneselam bamboo.

Factors	Comparison			Tukey’s Method			
		Coeffi.	Std. Error	t	α>/t/	95% Confid. Interval	
Ages (Yrs)	2 and 1	1.1	0.918	1.2	0.464	−1.18	3.38
	3 and 1	−3	0.918	−3.27	0.008	−5.28	−0.72
	3 and 2	−4.1	0.918	−4.47	0.000	−6.38	−1.82
Months	2 and 11	0.667	0.977	0.68	0.501	−1.335	2.668

## Data Availability

All the required data is available within the article.

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
