# Peer review of "Influence of Age and Harvesting Season on The Tensile Strength of Bamboo-Fibre-Reinforced Epoxy Composites"

_materials, 2022, doi:10.3390/ma15124144_

Round 1
Reviewer 1 Report
This manuscript presents an investigation of the mechanical properties of bamboo and bamboo epoxy composites. Particularly, the influences of harvesting age, time and place are investigated.
Bamboo is a material of interest for many structural and semi-structural applications. For this reason, I find the results interesting, and they should be published.
My only concern with the manuscript is that it requires a linguistic and stylistic polishing. There are, e.g., typos and layout issues that should be considered. In the attached file many of them are marked, but there are more.

Author Response
Title: Influence of Ages and Harvesting Season on The Tensile Strength of Bamboo Fibre Reinforced Epoxy Composites
MATERIALS-ID-1759205
The authors are highly grateful to the reviewers for their constructive comments, which have helped to enhance the quality of the manuscript. Author's sincere effort has been put to revise the manuscript according to the reviewer comments. The details of revisions made in response to the comments are summarized in the following Table. The revisions in the manuscript are shown in YELLOW colored texts.
|
|
|
|
REVIEWER-COMMENT-1 |
AUTHOR RESPONSE |
|
Comment; This manuscript presents an investigation of the mechanical properties of bamboo and bamboo epoxy composites. Particularly, the influences of harvesting age, time and place are investigated. Bamboo is a material of interest for many structural and semi-structural applications. For this reason, I find the results interesting, and they should be published. My only concern with the manuscript is that it requires a linguistic and stylistic polishing. There are, e.g., typos and layout issues that should be considered. In the attached file many of them are marked, but there are more.
|
Reply: The authors appreciate the reviewer for your valuable comments on typos, layout styles to improve the quality of the manuscript in all aspects. To address the issues we thoroughly check and edit the manuscript. Bamboo species in Ethiopia have not been investigated so far. The current research studies measure the mechanical strength of bamboo fibres and their epoxy composites based on harvesting age, seasons, and places. Yes, the properties of bamboo fibres are interesting because they replace glass fibres. According to the current results, the age, harvesting season, and type of bamboo species influenced the properties of bamboo fibres and their composites. |
|
|
|

Reviewer 2 Report
The current paper systematically studied the influence of ages and harvesting season on tensile strength of bamboo fibre reinforced epoxy composites. Some experimental data have certain significance to guide the engineering application of bamboo fiber composites. However, there are some non-standard writing problems in the current papers, for example, figures, tables and equations and so on. In addition, some other necessary supplements should be added based on the following major comments.
1# In the abstract, please add the effect mechanism of age and harvest season on the tensile strength and tensile modulus of bamboo fiber and composites at three locations.
2# The introduction part should be divided into several paragraphs for writing. The current introduction contains only one paragraph, which is very difficult for readers to read and understand.
3# According to the type of fiber, it can be divided into synthetic fiber (for example, carbon fiber, glass fiber and basalt fiber) and natural fiber (ramie fiber, flax fiber, bamboo fiber). As known, synthetic fiber reinforced polymer composites have been widely used in structures and engineering, such as the strengthening/reinforcing/repairing of concrete/steel members, strengthening steel members owing to the light weight, high strength, excellent corrosion resistance, fatigue resistance and creep resistance. Without the comparison, the analysis and summary on the advantages of natural fiber polymer composites is abrupt. Please refer to the following properties and advantages of synthetic fiber reinforced polymer composites for relevant supplement. Mechanical properties: https://doi.org/10.1080/15376494.2021.1974620. Fatigue performance: Materials & Design, 2019, 183: 108112. Durability: Construction and Building Materials, 2022, 315: 125710.
4# The interface bonding properties between bamboo fiber and polymer are poor. Therefore, are there any relevant references to carry out surface treatment to improve the interface performance between fiber and resin? In the last paragraph of the introduction, the authors should further highlight and emphasize the innovation of the current research work.
5# In part 2.1, Why do you choose bamboo species from three regions of Ethiopia? Are they typical? Are the properties of bamboo fiber related to region, climate and environment?
6# The age of bamboo fiber selected by the author is 1 ~ 3 years. Please indicate the basis for this choice.
7# Please improve the clarity of all pictures. For example, figure 1 and Figure 2 have been deformed and the clarity is very low. Figure 3 has obscured the name of Figure 3. All equations should be written in accordance with the specifications.
8# The writing logic of current materials and methods is chaotic. It is suggested to carry out them according to the following parts: the fiber extraction process and tests, epoxy preparation process, composite material preparation process and test methods.
9# From part 3.1-3.3, it can be found that the tensile property of bamboo fiber is the best in two years. Please explain the reason. In addition, it can also be found that the tensile property of bamboo fiber is related to regional location. In contrast, it has little to do with the harvest season. Why?
10# It can be seen from figure 8 that there is a large gap between the experimental data and the theoretical value of the young’s modulus of the materials. Please explain this.
11# From the whole test data, it can be found that the standard deviation of the tensile property data of the composite is large. Does this mean the instability of the sample during the preparation process?
12# It is suggested to divide the conclusion into 3 ~ 4 key points for writing.
Author Response
Title: Influence of Ages and Harvesting Season on The Tensile Strength of Bamboo Fibre Reinforced Epoxy Composites
MATERIALS-ID-1759205
The authors are highly grateful to the reviewers for their constructive comments, which have helped to enhance the quality of the manuscript. Author's sincere effort has been put to revise the manuscript according to the reviewer comments. The details of revisions made in response to the comments are summarized in the following Table. The revisions in the manuscript are shown in YELLOW colored texts.
|
|
|
|
REVIEWER-2-COMMENTS |
AUTHOR RESPONSE |
|
1# In the abstract, please add the effect mechanism of age and harvest season on the tensile strength and tensile modulus of bamboo fiber and composites at three locations. |
Reply: Yes, thank you for your valuable comments, we add the effect of age, harvesting seasons on the tensile strength and tensile modulus on the three location of bamboo species. The highest and the lowest tensile strength and tensile modulus of Kombolcha and Mekaneselam bamboo were recorded at the ages of 2 and 3 years old in the harvesting seasons of February and November, respectively. In contrast, the highest and the lowest tensile strength and tensile modulus of Injibara bamboo were recorded at the ages of 2 and 3 years old in February. However, the highest and lowest tensile strength and tensile modulus were recorded at the ages of 2 and 1 years old in November. The tensile strength and tensile modulus of harvesting month of November has higher value recorded compared to February. |
|
2# The introduction part should be divided into several paragraphs for writing. The current introduction contains only one paragraph, which is very difficult for readers to read and understand. |
Reply: Yes, we agree with the valuable comment by the reviewer: We arranged and divided the introduction section into paragraph in the manuscript. |
|
3# According to the type of fiber, it can be divided into synthetic fiber (for example, carbon fiber, glass fiber and basalt fiber) and natural fiber (ramie fiber, flax fiber, bamboo fiber). As known, synthetic fiber reinforced polymer composites have been widely used in structures and engineering, such as the strengthening/reinforcing/repairing of concrete/steel members, strengthening steel members owing to the light weight, high strength, excellent corrosion resistance, fatigue resistance and creep resistance. Without the comparison, the analysis and summary on the advantages of natural fiber polymer composites is abrupt. Please refer to the following properties and advantages of synthetic fiber reinforced polymer composites for relevant supplement. Mechanical properties: https://doi.org/10.1080/15376494.2021.1974620. Fatigue performance: Materials & Design, 2019, 183: 108112. Durability: Construction and Building Materials, 2022, 315: 125710. |
Reply: Thank you for your suggestion and for giving us a link for further knowledge, then we took some content and used it as a reference. |
|
4# The interface bonding properties between bamboo fiber and polymer are poor. Therefore, are there any relevant references to carry out surface treatment to improve the interface performance between fiber and resin? In the last paragraph of the introduction, the authors should further highlight and emphasize the innovation of the current research work. |
Reply: Thank you, the authors appreciate the reviewer for your valuable comments on the introduction parts that we should further highlight and emphasize the innovation of the current research work. We accepted and incorporated as per your comments. The matrix of the current research studies is epoxy, which is compatible with natural fibres compared to thermoplastics matrix. Yes, we can add relevant references about the surface treatment of fibres to improve compatibility, but the performance of interfacial bonding is not part of the current research topic. |
|
5# In part 2.1, Why do you choose bamboo species from three regions of Ethiopia? Are they typical? Are the properties of bamboo fiber related to region, climate and environment? |
Reply: Thank you for your questions, the properties of Ethiopian bamboo species has not been investigated by researchers so far. The aim of the current research studies is to measure the strength of Ethiopian bamboo fibres and their epoxy composites based on age and harvesting season. As we know, the properties of natural fibres are influenced by age, climate, type of species, environment, etc. They have no consistent properties. Yes, the bamboo species are typical in their regions, so the current research studies are based on regions. |
|
6# The age of bamboo fiber selected by the author is 1 ~ 3 years. Please indicate the basis for this choice. |
Reply: Yes, the authors are grateful for the valuable suggestion, we indicate the basis of choice of the age as per your suggestion. The choice of age is indicated and cited on the manuscript of the literature review. The bamboo plants matured at the ages of 3 to 4. Since the mature period of bamboo plants is in the range of 3 to 4 years, the current research work is selected for the ages of 1, 2, and 3 years. |
|
7# Please improve the clarity of all pictures. For example, figure 1 and Figure 2 have been deformed and the clarity is very low. Figure 3 has obscured the name of Figure 3. All equations should be written in accordance with the specifications. |
Reply: We agree with the valuable comment by reviewer to improve the clarity of Fig. 1, Fig. 2, Fig. 3, and we prepare the equation in accordance with the specification. |
|
8# The writing logic of current materials and methods is chaotic. It is suggested to carry out them according to the following parts: the fiber extraction process and tests, epoxy preparation process, composite material preparation process and test methods. |
Reply: We sincerely appreciate the reviewer for the valuable comment, we prepare as per your suggestion. |
|
9# From part 3.1-3.3, it can be found that the tensile property of bamboo fiber is the best in two years. Please explain the reason. In addition, it can also be found that the tensile property of bamboo fiber is related to regional location. In contrast, it has little to do with the harvest season. Why? |
Reply: Thank you for your question. Yes, the properties of bamboo fibres are best at the age of 2 years, because the thickening growth of cell walls in bamboo fibers near the outer surface of bamboo is almost completed by 2 years. The complete cell wall maturation of bamboo plants occur at the age of 2 years. Seasons in Ethiopia are categorized into 3 periods, such as rainy (June-September), cold (October-January), and dry (February-May). The aim of the current research studies is to characterize the influence of the harvesting season on the properties of bamboo fibres. The authors chose two harvesting seasons for the current research due to the limitation of time. |
|
10# It can be seen from figure 8 that there is a large gap between the experimental data and the theoretical value of the young’s modulus of the materials. Please explain this. |
Reply: Thank for your question: The gap in the percentage of theoretical and experimental Young's modulus is less than 5%. So it is a good, close-value relationship between them. This value indicates that it is over 95% confidence value of good preparation of the specimen. |
|
11# From the whole test data, it can be found that the standard deviation of the tensile property data of the composite is large. Does this mean the instability of the sample during the preparation process? |
Reply, I would like to appreciate your question: The cause of standard deviation comes from three sources: the specimen preparation process, the testing process, and the accuracy of the testing machine. The standard deviation may be largely due to the above causes. The current research studies of standard deviation are in the range of 10%–15% of its value, which has a good deviation because it is coming from many sources of events |
|
12# It is suggested to divide the conclusion into 3 ~ 4 key points for writing. |
Reply: The authors appreciate the reviewer for your valuable suggestions on conclusions to improve the quality of the manuscript in all aspects. |
|
|
|

Round 2
Reviewer 2 Report
It can be accepted the paper.